# Ferroelectric columnar assemblies from the bowl-to-bowl inversion of aromatic cores

Shunsuke Furukawa [1✉], Jianyun Wu[2], Masaya Koyama[1], Keisuke Hayashi[1], Norihisa Hoshino [2,3],
Takashi Takeda [2,3], Yasutaka Suzuki[4], Jun Kawamata[4], Masaichi Saito [1✉] & Tomoyuki Akutagawa [2,3✉]

Organic ferroelectrics, in which the constituent molecules retain remanent polarization, represent an important topic in condensed-matter science, and their attractive properties, which include lightness, flexibility, and non-toxicity, are of potential use in state-of-the-art ferroelectric devices. However, the mechanisms for the generation of ferroelectricity in such organic compounds remain limited to a few representative concepts, which has hitherto severely hampered progress in this area. Here, we demonstrate that a bowl-to-bowl inversion of a relatively small organic molecule with a bowl-shaped π-aromatic core generates ferroelectric dipole relaxation. The present results thus reveal an unprecedented concept to produce ferroelectricity in small organic molecules, which can be expected to strongly impact materials science.

[1] Department of Chemistry, Graduate School of Science and Engineering, Saitama University, Shimo-okubo, Sakura-ku, Saitama-city, Saitama 338-8570, Japan. [2] Graduate School of Engineering, Tohoku University, Sendai 980-8579, Japan. [3] Institute of Multidisciplinary Research for Advanced Materials (IMRAM), Tohoku University, 2-1-1 Katahira, Aoba-ku, Sendai 980-8577, Japan. [4] Graduate School of Sciences and Technology for Innovation, Yamaguchi University, 1677-1 Yoshida, Yamaguchi 753-8512, Japan. ✉email: furukawa@mail.saitama-u.ac.jp; masaichi@chem.saitama-u.ac.jp; akutagawa@tohoku.ac.jp

Ferroelectric materials, which undergo reversible electric polarization upon exposure to an external electric field, are crucial components of ferroelectric devices, such as non-volatile memories, actuators, and piezoelectric devices[1]. The ferroelectricity of organic molecules has attracted attention as a promising alternative to inorganic ferroelectric materials that contain precious and/or toxic metals[2,3]. However, examples of organic ferroelectrics are relatively scarce, and those known typically present one of the following three mechanisms: (1) orientational change of polar low-molecular-weight compounds[4–6], (2) electron transfer in donor–acceptor-type charge-transfer complexes[7–9], and (3) proton transfer in hydrogen-bonding networks[10–12] (Fig. 1, left). Therefore, the development of new strategies for designing organic ferroelectrics is of fundamental importance and represents a primordial step for versatile memory applications. Chemical design strategies based on π-electron systems afford control over electrical conduction, as well as magnetic and optical properties. For instance, the charge-ordered state of the cation radical salts of bis(ethylenedithio)tetrathiafulvalene leads to electronic polarization-type ferroelectrics[13]. On the contrary, both the flexibility and the dynamics of molecular assemblies of low-molecular-weight compounds offer an interesting dynamic environment that may allow controlling certain physical properties, such as dislocation, ferroelectricity, and ferroelasticity. Indeed, ferroelectric responses have been observed in the flip-flop motion of a supramolecular cation and the multi-axial rotation of polar molecules in a plastic crystalline state[6,14]. Such diverse motional freedom in molecular assemblies could potentially be exploited to fabricate functional ferroelectric molecular memory materials. The supramolecular assembly of liquid crystalline compounds in a bowl-shaped structure induces dipole moments that subsequently lead to ferroelectricity (Fig. 1, middle)[15,16]. Although bowl inversions have typically been observed in solution, ferroelectricity that originates from bowl inversion in supramolecular assemblies has not yet been realized, even though some advances using bowl-shaped molecules have been reported[17–19]. For example, subphthalocyanine derivatives with strong axial dipoles have shown permanent homeotropic alignment and polarization in columnar liquid crystals with electric fields[20] and ferroelectric properties in

nematic phases[21]. On the other hand, the use of polar oligo-(vinylidenedifluoride) and/or amide side chains are also useful to induce ferroelectricity in solid-state columnar organic π-conjugated molecules[22,23]. A bowl-to-bowl inversion mechanism for the ferroelectric response is hardly designed in solid-state columnar assembly, necessitating the utilization of highly thermal fluctuated liquid crystal or plastic crystal states.

Here, we report organic ferroelectric materials based on hexaalkoxy trithiasumanenes (**CnSS**), which exhibit invertible bowl-shaped π-aromatic cores. We discovered that the bowl-shaped aromatic cores of these **CnSS** molecules are one-dimensionally stacked in a columnar fashion, and that the dipole moment originating from the aligned bowl-shaped framework leads to remanent polarization in the crystals. The present work thus describes a concept of invertible aromatic cores for materials design that provides facile access to solution-processable and low-cost single-columnar ferroelectric memory devices with a high density (>12 TB cm$^{-2}$).

## Results

**Molecular design and synthesis.** We selected trithiasumanene (**SS**)[24] as the bowl-shaped π-aromatic core, given its low bowl-inversion barrier. In general, the magnitude of the bowl-inversion barrier ($\Delta E_b$) is closely related to the bowl depth ($l_b$) of the curved π-frameworks. For the typical bowl-shaped molecule sumanene ($l_b = 1.143$ Å; $\Delta E_b = 18.2$ kcal mol$^{-1}$)[25], these parameters are higher than those of corannulene ($l_b = 0.877$ Å and $\Delta E_b = 9.2$ kcal mol$^{-1}$)[26] (Fig. 2). The introduction of large heteroatoms, such as sulfur and selenium in curved π-molecules affords heterasumanenes, whereby the dynamic and electronic properties can be tuned by judicious choice of the inserted heteroatom(s)[27–31]. Among these, **SS** molecules that bear three S atoms in the sumanene π-skeleton exhibit even smaller values ($l_b = 0.76$ Å and $\Delta E_b = 1.9$ kcal mol$^{-1}$) than those of sumanene and corannulene[25]. The large atomic radius of the S-atom leads to a reduction of both parameters for the π-framework, whereas Se-atoms, which are even larger, afford the almost planar triselenasumanenes (**SSe**) (Supplementary Fig. 1). The shallower bowl depth of **SS** should facilitate the bowl-to-bowl inversion of the

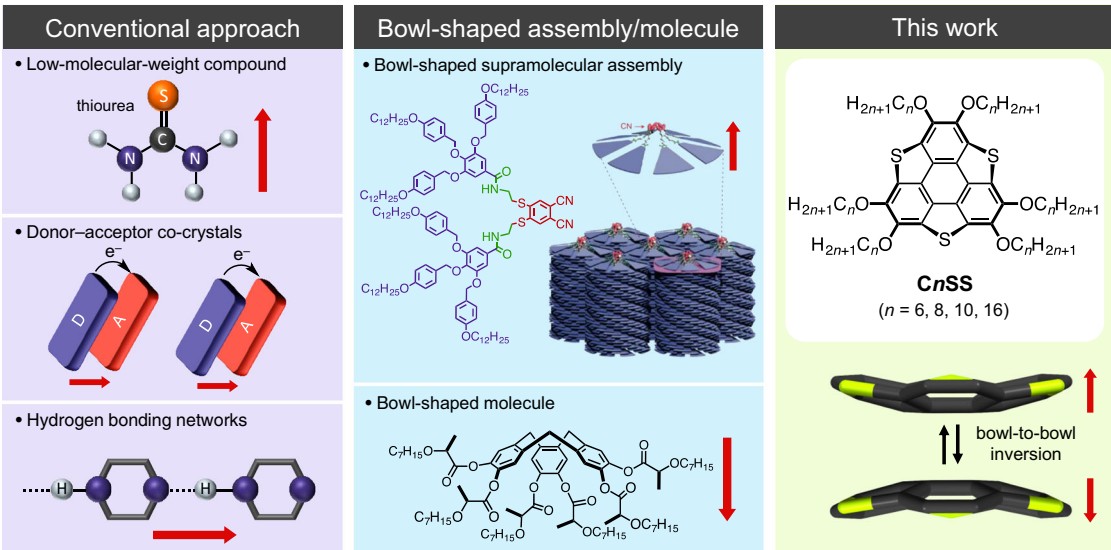

**Fig. 1 Conventional approaches towards organic ferroelectric materials: anisotropic low-molecular-weight compounds, donor–acceptor-type charge-transfer complexes, and proton transfer in hydrogen-bonding networks (left).** A ferroelectric bowl-shaped supramolecular assembly (reprinted from ref. [3]) and the chemical structure of a bowl-shaped molecule (middle). Chemical structures of **CnSS** molecules that contain a bowl-to-bowl invertible π-core (right). The red arrows indicate the dipole moment in each system.

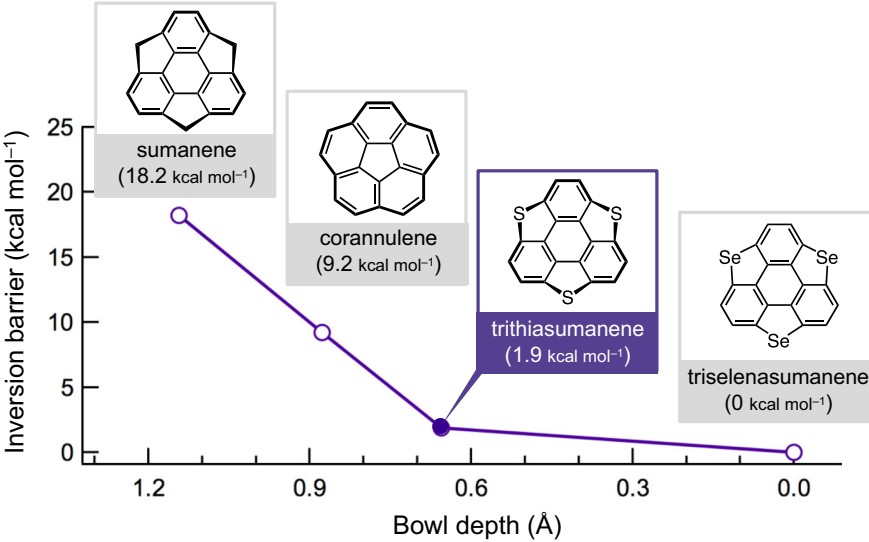

**Fig. 2** Correlation between bowl depth and inversion barrier in bowl-shaped π-conjugated molecules.

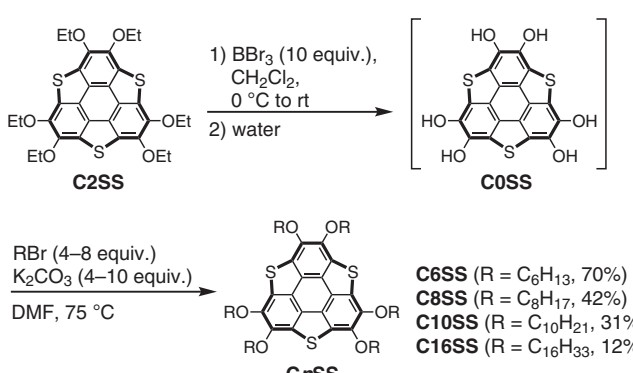

**Fig. 3** Synthesis of the targeted **C$n$SS** derivatives ($n$ = 6, 8, 10, or 16).

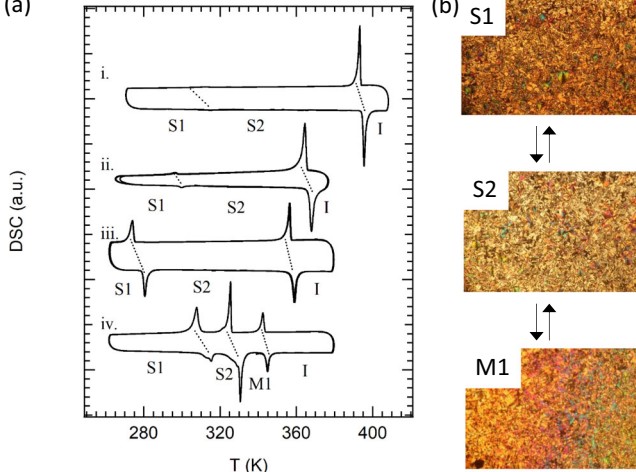

**Fig. 4 Phase-transition behavior of C$n$SS derivatives ($n$ = 6, 8, 10, or 16).**
**a** DSC curves of (i) **C6SS**, (ii) **C8SS**, (iii) **C10SS**, and (iv) **C16SS**, where transformations between the S1, S2, and M1 phases were reversible in the $T$-cycle. **b** POM images of **C16SS** at S1 ($T$ = 300 K), S2 ($T$ = 320 K), and M1 ($T$ = 330 K).

aromatic core in the solid state upon applying an external stimulus. To promote motional freedom, as well as bowl inversion, six long alkoxy chains ($-OC_nH_{2n+1}$) were introduced in the **SS** π-core, as such motional freedom should enhance the flexibility and internal thermal energy of the molecular assembly upon heating.

Herein, we report the synthesis of **SS** derivatives that bear $-OC_nH_{2n+1}$ chains (**C$n$SS**; $n$ = 6, 8, 10, or 16) by dealkylation of **C2SS**, followed by re-alkylation of the obtained hexahydroxy intermediate **C0SS** (Fig. 3). The starting material (**C2SS**) was prepared by a modified method for the synthesis of **C4SS**[32]. Treatment of **C2SS** with boron tribromide, followed by addition of water, afforded the air-sensitive hydroxy intermediate **C0SS**, which was treated with the corresponding alkyl bromides ($C_nH_{2n+1}$–Br; $n$ = 6, 8, 10, or 16) and potassium carbonate in dimethylformamide (DMF) to obtain the **C$n$SS** derivatives.

**Phase-transition behavior and molecular assembly**. The differential scanning calorimetry curves of **C$n$SS** ($n$ = 6, 8, 10, or 16) in the solid state show a reversible phase transition (Fig. 4a). Crystals of **C6SS**, **C8SS**, and **C10SS** exhibit one reversible solid–solid (S1–S2) phase transition at 315, 299, and 280 K upon heating, and at 313, 297, and 274 K upon cooling, with melting points (S2–I phase transition) at 395, 368, and 359 K, respectively (Table 1). The increasing length of the $-OC_nH_{2n+1}$ chains results in lower S1–S2 and S2–I phase-transition temperatures. In polarized optical microscopy (POM) images, both the S1 and S2 phases exhibit birefringence under a cross-Nicol optical alignment in the absence of fluid behavior (top and middle POM images in Fig. 4b). On the contrary, fluid and birefringence properties were confirmed for the M1 phase of **C16SS** (bottom POM image in Fig. 4b), suggesting the formation of a liquid crystalline phase before melting. The M1–I phase-transition temperature and transition-enthalpy change ($\Delta H$) for **C16SS** are 345 K and 15.1 kJ mol$^{-1}$, respectively. The high structural flexibility of the $-OC_{16}H_{33}$ chains becomes apparent in the liquid crystalline phase above the S2 phase (330–345 K). The phase-transition behavior of the **C$n$SS** derivatives is very similar, except for the stabilization of the M1 phase for the **C16SS** derivative.

Next, we determined the molecular and assembly structure of **C$n$SS** ($n$ = 6, 8, 10, or 16) by X-ray diffraction analyses. Although a single-crystal X-ray diffraction analysis was successfully carried

 3

**Table 1 Summary of structural features, thermal properties, and *P–E* hysteresis of C*n*SS, C*n*SeS, and C6TP.**

| Compound | Structure of aromatic core | Phase-transition and melting temperature[d] (K) | | | | *P–E* response |
|---|---|---|---|---|---|---|
| | | S1–S2 | S2–M1 | S1– Col$_h$ | S2/M1/Col$_h$–I | |
| **C6SS**[a] | Bowl | 315 | – | – | 395 | Hysteresis |
| **C8SS**[a] | Bowl | 299 | – | – | 368 | Hysteresis |
| **C10SS**[b] | Bowl | 280 | – | – | 359 | Hysteresis |
| **C16SS**[c] | Bowl | 315 | 330 | – | 345 | Hysteresis |
| **C4SeS**[b] | Planar/bowl[d] | 303 | – | – | 353 | Linear |
| **C6TP**[c] | Planar | – | – | 343 | 371 | Linear |

[a]Obtained at heating process of the DSC analysis.
[b]Absense of fluid behavior in S1 and S2 phases.
[c]Liquid crystalline state at M1 or Col$_h$ phase.
[d]Structure of triselenasumanene core of C4SeS obtained by crystallographic analysis is bowl-shaped, whereas the optimized structure obtained by the theoretical calculation is planar.

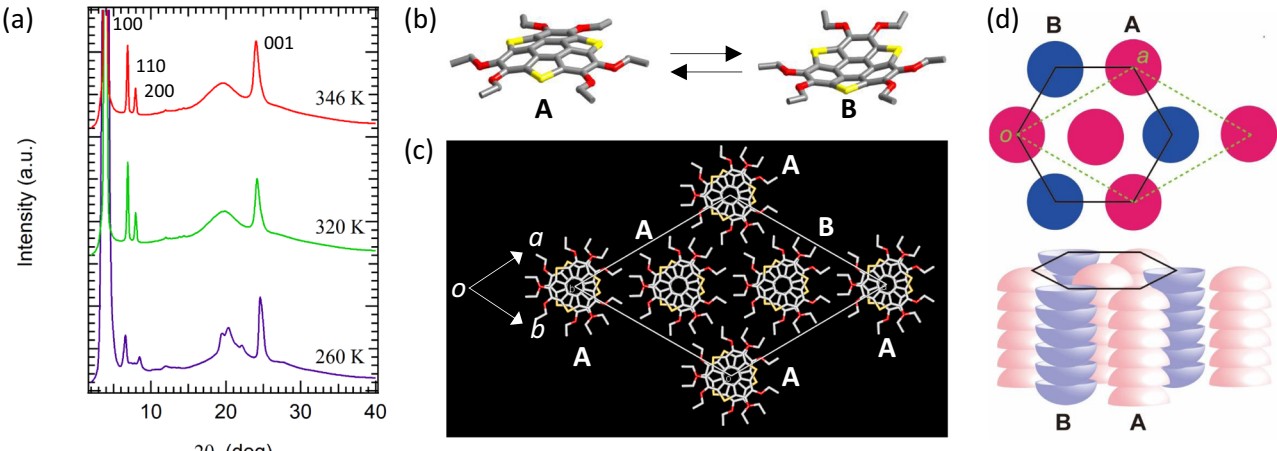

**Fig. 5 Molecular assembly structure of C*n*SS. a** PXRD patterns at 260 K (S1 phase), 320 K (S2 phase), and 346 K (S2 phase) with index assignments for the 100, 110, 200, and 001 reflections. **b** Bowl-to-bowl inversion from the A (up) to the B (down) conformation. **c** Unit cell of **C2SS** (viewed along the *c*-axis) based on the single-crystal X-ray diffraction analysis at 300 K. The A and B columns coexist in the unit cell with an occupation ratio of A/B = 2/1, suggesting a permanent dipole moment along the *c*-axis. **d** Schematic arrangement of the A and B columns in the hexagonal columnar (black solid lines) and trigonal (green dotted lines) lattices. The A and B columns are shown as pink and blue bowls, respectively.

out for **C2SS** at 300 K, the single crystals obtained for the long-chain derivatives (*n* = 6, 8, 10, or 16) were unfortunately of insufficient quality. Thus, a temperature-dependent powder X-ray diffraction (PXRD) analysis was performed for each S1, S2, and M1 phase to obtain information on the molecular assembly structures. Figure 5a shows the temperature-dependent PXRD patterns for the S1 (*T* = 260 K) and S2 (*T* = 320 and 346 K) phases of **C10SS**. **C6SS** and **C8SS** exhibit almost identical PXRD patterns (Supplementary Fig. 2). The pattern of the S1 phase of **C10SS** (purple in Fig. 5a) is similar to that of the S2 phase. However, several sharp diffraction peaks at $2\theta \sim 20°$ broadened upon transition to the S2 phase (red and green in Fig. 5a), suggesting that the crystal symmetry of the latter is probably higher than that of the S1 phase due to thermally activated molecular motion. The most intense diffraction peak of the S2 phase at $2\theta = 3.960°$ was assigned to the 100 reflection with a periodicity of $a = 47.82$ Å for the trigonal (or hexagonal) crystal system, which corresponds to $a = 25.4783(2)$ Å for the single-crystal X-ray diffraction analysis of **C2SS** with the polar space group *P*3*c*1 at 300 K. In addition, the two weak diffraction peaks at $2\theta = 6.908$ and $7.964°$ were assigned to the 110 and 200 reflections, respectively, which is consistent with the formation of a hexagonal lattice. A linear relationship of the *h*00 reflections was confirmed in the **C*n*SS** series (*n* = 2, 6, 8, 10, or 16)

(Supplementary Fig. 3), suggesting an identical packing structure for all **C*n*SS** derivatives. The broad reflection at $2\theta \sim 20°$ is probably due to the melting of the $-OC_nH_{2n+1}$ chains, similar to the behavior in a discotic columnar liquid crystalline phase. It should be noted that the intense reflection at $2\theta \sim 24.02°$ corresponds to an average π-stacking distance of 3.71 Å.

A single-crystal X-ray diffraction analysis of **C2SS** at 300 K provided insight into the solid-state structure of **C2SS**, and the principal arrangements can potentially be extrapolated to the other **C*n*SS** derivatives with longer alkyl chains. The molecular and packing structures for **C2SS** based on the single-crystal X-ray diffraction analysis are shown in Fig. 5b, c, respectively. A bowl-shaped molecular structure was confirmed, where both the up-bowl (A) and down-bowl (B) columns coexist in the unit cell. The π-stacking columns of A and B are hexagonally arranged with respect to each other (Fig. 5c). The occupation ratio of A-column/B-column (2/1) in the unit cell is consistent with the formation of the polar space group *P*3*c*1. Figure 5d shows the schematic arrangement of π-column A (red) and B (blue) in the trigonal (dashed green line) and hexagonal (solid black line) lattices. The polar space group *P*3*c*1 of **C2SS** is consistent with a ferroelectric ground state along the *c*-axis, assuming bowl-to-bowl inversion between A and B. The PXRD patterns of the S1 and S2 phases for liquid crystalline **C16SS** are very similar, and that of the M1

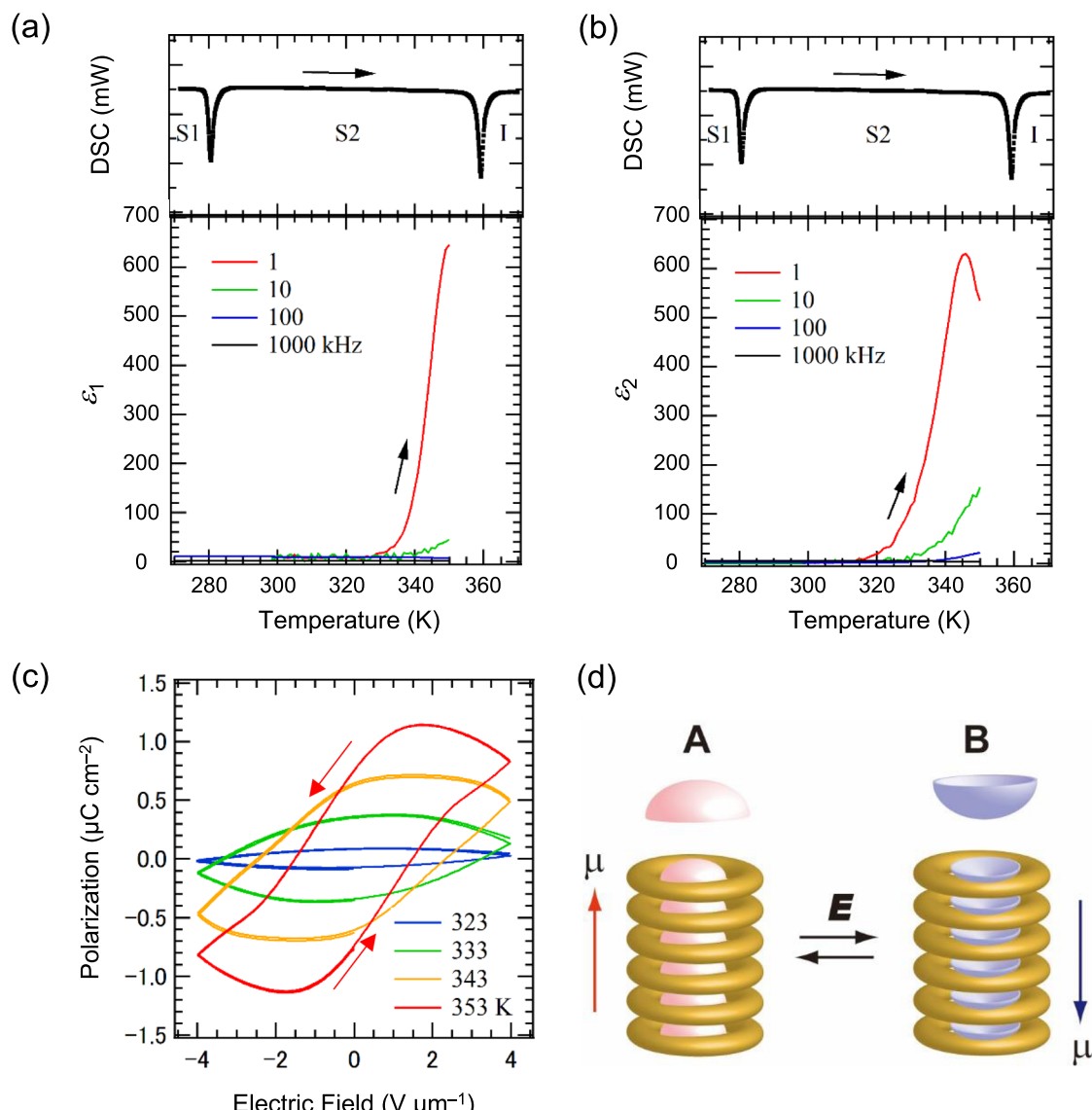

**Fig. 6 Dielectric properties of C10SS.** *T*- and *f*-dependent **a** real $\varepsilon_1$ and **b** imaginary $\varepsilon_2$ components together with the DSC curves during the heating process. **c** *T*-dependent *P–E* hysteresis curves at *f* = 100 Hz. **d** Schematic model of the bowl-to-bowl structural inversion upon application of an electric field, where molten $-OC_nH_{2n+1}$ chains surround the polar bowl column.

phase is consistent with the formation of a discotic hexagonal columnar ($Col_h$) liquid crystal phase (Supplementary Fig. 4b). In this $Col_h$ phase, the thermally activated melting of the six $-OC_{16}H_{33}$ chains and the free rotation of the π-stacking columns along the orientation of the director occur simultaneously, weakening the intensity of the broad 001 reflection due to thermal fluctuation of the π-stacking $d_{001}$-spacing. Although the $Col_h$ mesophase was not observed for **C6SS**, **C8SS**, and **C10SS**, the molten state of the six $-OC_nH_{2n+1}$ chains was confirmed by the broad diffraction at $2\theta \sim 20°$ (Supplementary Fig. 2). The in-plane rotation of each bowl-shaped molecule is likely suppressed in the π-stacking columns of the S2 phase, also disappearing in the $Col_h$ phase for certain **CnSS** (*n* = 2, 6, 8, or 10). The thermally activated molten state of the $-OC_nH_{2n+1}$ chains plays an important role in the phase transitions and in the ferroelectric response of such bowl-shaped **CnSS**.

**Ferroelectric response**. Dielectric features arise from the molecular motion of polar structural units within a molecular assembly. In the present **CnSS** derivatives, such polar structural units are only observed in the bowl-shaped π-plane, whose π-stacking generates a macro dipole moment along the π-stacked column. The π-stacked columns of **CnSS** in high-temperature S2 phase were easily oriented along the direction normal to the ITO surface after the cooling from isotropic liquid, which was confirmed in the dark POM images of the homeotropic orientation on the substrate surface (Supplementary Fig. 5). A polar crystal structure with space group *P3c1* was identified for a single crystal of **C2SS**, where the occupation number of the polar A and B π-stacking bowls was not balanced, producing a ferroelectric dipole ground state. The second-harmonic generation (SHG) activity was observed at **C2SS** at 298 K, in which the magnitude was similar to that of sucrose (Supplementary Figs. 6–7). Figure 6a, b shows the temperature (*T*)- and frequency (*f*)-dependent real ($\varepsilon_1$) and imaginary ($\varepsilon_2$) components of the dielectric constants of **C10SS** using a sandwich-type electrode. The *T*- and *f*-dependent $\varepsilon_1$ responses around 300 K increased monotonically with increasing temperature and frequency above 330 K for the S2 phase. Dielectric anomalies were not observed around the S1–S2 phase transition. On the contrary, the $\varepsilon_2$–*T* plots at *f* = 1 kHz

showed a dielectric peak at 349 K, i.e., a lower temperature than that of the S2–I phase transition (356 K). Almost the same $f$- and $T$-dependent dielectric constants were obtained for **C6SS** and **C8SS** (Supplementary Fig. 8). Interestingly, a ferroelectric polarization–electric field ($P$–$E$) hysteresis at $f = 100$ Hz for **C10SS** was confirmed in the high-temperature S2 phase (Fig. 6c), i.e., it was clearly observed at 353 and 343 K with a remanent polarization ($P_r = 0.5$–$0.7$ μC cm$^{-2}$) and coercive voltage ($E_{th} \approx 1.5$ V μm$^{-1}$). Time-dependent polarization behavior of **C10SS** at 343 K showed that the polarization was retained for 4000 ms after applying a pulse voltage (Supplementary Fig. 9). The appearance of an $\varepsilon_2$ peak at 356 K and $f = 100$ Hz is consistent with the thermally activated molecular motion of the polar structural units at ~100 Hz and 356 K, corresponding to a structural inversion of the π-stacking columns (Fig. 6d). The observed $P$–$E$ hysteresis curve is explained by the polling process between the electrodes, which orients each column and domain along the sandwich direction. The homeotropic orientation of each π-stacking column aligns the direction of macro dipole moment normal to the ITO surface without the polling process. The bowl-to-bowl structural inversion is thermally activated even in the solid S2 phase. The thermal melting of the six $-OC_nH_{2n+1}$ chains around the polar π-stacking column plays an essential role in the dipole inversion of each polar column, resulting in ferroelectric dipole relaxation via structural bowl-to-bowl inversion. Similar ferroelectric relaxation processes were observed in the S2 phase of **C6SS** and **C8SS** (Supplementary Fig. 10), whereas liquid crystalline **C16SS** also exhibited ferroelectricity in the M1 phase ($E_{th} \approx 2.2$ V μm$^{-1}$; $P_r \approx 0.4$ μC cm$^{-2}$; $T = 336$ K) (Supplementary Fig. 11).

## Discussion

The shallower bowl depth of the **SS** π-core is crucial for the bowl inversion in the solid state. The dipole moment ($\mu$) along the bowl-shaped π-plane of **SS** is 1.03 D[25]. The theoretically calculated bowl-to-bowl inversion energy ($\Delta E_b$) in the gas phase is relatively low (1.9 kcal mol$^{-1}$), i.e., by one order of magnitude lower than that of sumanene ($\Delta E_b = 18.2$ kcal mol$^{-1}$)[25] (Fig. 2), but higher than the room temperature energy ($k_B T \approx 0.58$ kcal mol$^{-1}$). The bowl-to-bowl inversion of **CnSS** ($n = 6$, 8, 10, or 16) was observed at ~350 K (~0.7 kcal mol$^{-1}$) in the π-stacked one-dimensional molecular assembly, whereas such an inversion could not be observed in the liquid crystalline phase of a $-SC_nH_{2n+1}$-substituted sumanene derivative[19]. The small curvature angle of the π-plane of **SS** leads to a smaller $\Delta E_b$, thus enabling the bowl-to-bowl inversion even in the solid state, where the thermal melting of the alkyl chains serves as an entropy source for the polar π-stacking structure.

In order to assess the correlation between the ferroelectricity and the molecular structure, the shallow bowl-shaped selenium derivative **C4SeS** was prepared by replacement of three S atoms by Se[32]. Moreover, the π-planar triphenylene derivative **C6TP** was evaluated in terms of its phase transitions, $T$- and $f$-dependent dielectric constants, and $P$–$E$ hysteresis curves. Although the theoretically obtained structure of triselenasumanene (**SeS**) is planar in the absence of alkoxy groups (Fig. 2), the structure of **C4SeS** presents a shallow bowl shape with bowl depths ranging from 0.32 Å to 0.46 Å, as determined by a single-crystal X-ray diffraction analysis[32]. Crystals of **C4SeS** exhibit a phase transition from crystalline solid (S1) to isotropic liquid (IL) at 358 K in the absence of the intermediate thermally activated S2 and/or Col$_h$ liquid crystalline phases found for the **CnSS** derivatives (Supplementary Fig. 12a). Single crystals of **C4SeS** with polar space group $R3$ present one-dimensional polar bowl-to-bowl π-stacking columns along the $c$-axis, where the macroscopic dipole moment

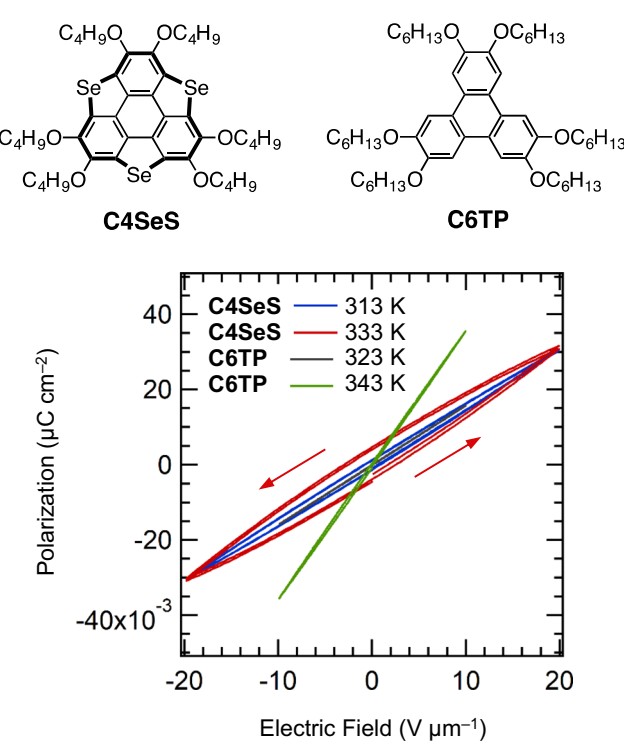

**Fig. 7** Molecular structures of **C4SeS** and **C6TP** (top) and $T$-dependent $P$–$E$ hysteresis curves at $f = 100$ Hz (bottom).

may potentially exhibit dipole inversion and ferroelectricity upon application of an external electric field, similar to the case of the **CnSS** crystals. It should be noted here that the crystalline S1 state of **C4SeS** does not contain thermally molten $-OC_nH_{2n+1}$ chains. The temperature-dependent PXRD patterns of **C4SeS** in the temperature range from 150 to 350 K show a highly crystalline phase with sharp diffraction peaks and the absence of a broad diffraction at $2\theta$~20°, which suggests a non-thermally activated molten state of the $-OC_4H_9$ chains (Supplementary Fig. 12b). Furthermore, dielectric anomalies were not observed for the $T$- and $f$-dependent $\varepsilon_1$-values, which is consistent with the absence of a bowl-to-bowl inversion in the crystalline S1 state (Supplementary Fig. 13). The $P$–$E$ curves of the S1 phase of **C4SeS** at 313 and 333 K revealed a linear behavior at $f = 100$ Hz, which is consistent with a paraelectric ground state (Fig. 7). Although a polar space group was observed for crystalline **C4SeS**, its highly crystalline and rigid molecular assembly structure suppresses the thermally activated bowl-to-bowl inversion. On the contrary, the π-planar molecule **C6TP** exhibits an S1–Col$_h$ phase transition at 343 K and a Col$_h$–I transition at 371 K, where a thermally stable columnar liquid crystalline phase was observed within a temperature range of ~30 K (Supplementary Fig. 14a). However, both the $T$- and $f$-dependent $\varepsilon_1$ behavior of **C6TP** did not show any evidence for bowl-to-bowl inversion or thermally activated molecular motion of the polar structural units within the molecular assembly (Supplementary Fig. 14b, c). Stacked structures of π-planar molecules do not exhibit a macroscale dipole moment. Therefore, linear $P$–$E$ hysteresis curves were obtained both in the S1 and Col$_h$ phases due to the absence of dipole inversion, despite the flexibility of the Col$_h$ phase. Therefore, thermally activated melting and/or a dynamic state of the peripheral $-OC_nH_{2n+1}$ chains and the bowl-to-bowl inversion at the central **SS** core are both essential for the ferroelectricity exhibited by the **CnSS** derivatives in the S2 and Col$_h$ phases.

To elucidate the ferroelectric mechanism of **C*n*SS**, the dipole moment and $P_r$ values were determined by DFT calculations at the B3LYP 6–31 G(d,p) level of theory on model compounds **SS** and **C1SS**. The macroscopic polarization, $P_r$, was obtained from Eq. (1):

$$P_r = \frac{\Sigma_{i=1}^n P_i}{\Delta V} = \frac{\Sigma p_{mol}}{\Delta V} \quad (1)$$

The calculated dipole moment along the normal to the bowl-shaped π-plane of the **SS** molecule ($\mu = 0.84$ D) afforded a theoretical $P_r$ value of 0.19 μC cm$^{-2}$ for **C*n*SS** at 343 K, which is inconsistent with the experimental value ($P_r \approx 0.6$ μC cm$^{-2}$). However, collective motion of dipole moment in the π-stacking column effectively enhanced the macroscale dipole moment and the magnitude of the polarization[33,34]. Although the SHG activity was clearly confirmed in highly crystalline **C2SS** with polar space

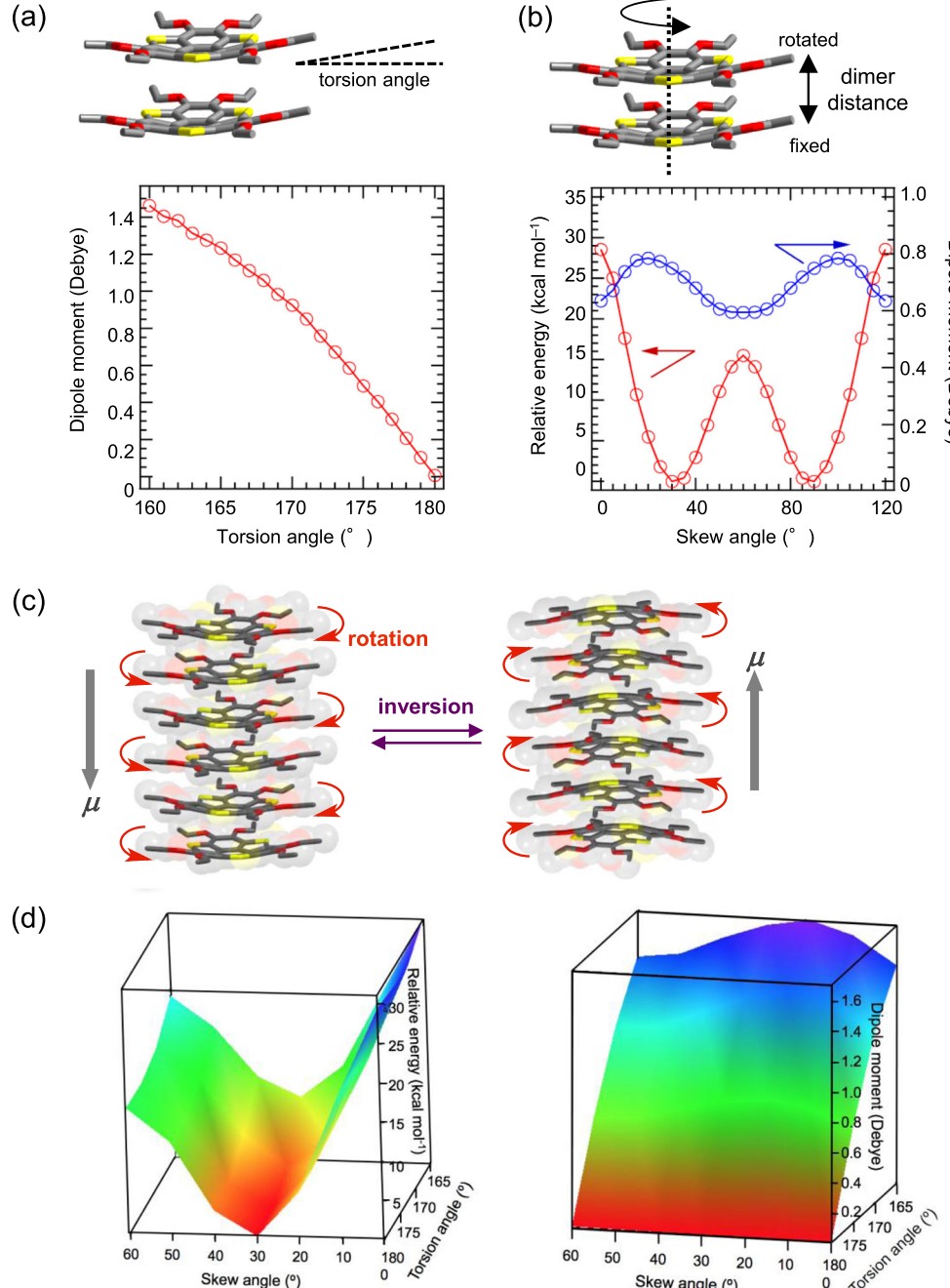

**Fig. 8 Ferroelectric relaxation mechanism of C*n*SS derivative. a** Relative energy (red in left scale) and dipole moment (blue in right scale) vs. torsion angel plots of dimeric (**C1SS**)₂ based on DFT calculations at the B3LYP 6-31 G(d) level of theory. The interdimer distance and skew angle between central benzene rings were fixed with those of **C2SS** dimer in the crystal. The smallest total energy of π-planar structure with a dihedral angle of 180° was defined as zero. **b** Relative energy (red in left scale) and dipole moment (blue in right scale) vs. skewed angel plots of dimeric (**C1SS**)₂. The total energy of bowl-shaped structure with a skew angle of 30° and 90° was defined as zero. **c** Schematic dipole inversion mechanism coupled with fluctuation/inversion of bowl-shaped conformation and skew rotation. **d** 3D plots of relative energy (left) and dipole moment (right) against torsion angle and skew angle of dimeric (**C1SS**)₂.

group of $P3c1$, there were insufficient SHG signals for the other ferroelectric **CnSS** derivatives at high-temperature S2 phase. The SHG activity should be observed in the ferroelectric ground state due to the polar space group, while highly viscous plastic crystalline S2 phase with thermally melting state of six $-OC_nH_{2n+1}$ chains exhibited the weak SHG activity in spite of their $P$-$E$ hysteresis. To discuss this ferroelectric relaxation behavior of **CnSS** derivatives, we estimated a possible polarization mechanism for dipole inversion and relaxation process of the bowl-to-bowl inversion based on the theoretical DFT calculation using a model compound of **C1SS** dimer (Fig. 8), where the interdimer distance and skew angle between central benzene rings were fixed at the same as those observed in the **C2SS** single-crystal X-ray analysis. The bowl-depth (torsion angle at C–C–C–S bonds) was increased in a stepwise manner from the π-planar structure (torsion angle = 180°) to the bowl-shaped one (torsion angle = 160°) of **C1SS** dimer (Fig. 8a). Thermally fluctuated structural transformation between the π-planar and bowl-shaped conformations at plastic crystalline S2 phase easily activated dipole moment from 0 D at π-planar structure to 1.46 D at bowl-shaped one by the increase in the torsion angle (Fig. 8a). The intermolecular interaction between the π-stacking columns was dominated by the hydrophobic interaction between the alkyl chains, and consequently thermal melting decreased the magnitude of intermolecular interaction and activated the bowl-to-bowl inversion in the column. Therefore, the thermally activated structural fluctuation between the polar and the non-polar molecular conformations in the 1D π-stacking column should be easily activated in highly temperature viscous S2 phase, where the polarity and SHG activity were thermally disturbed in bulk due to the thermal fluctuation.

The second important molecular motion is a relative skew rotation of **C1SS** dimer, where the rotation angle of the upper **C1SS** relative to the lower one changed the magnitude of the relative energy at fixed bowl-shaped conformation with the fixed torsion angle of 175° (Fig. 8b). When the bowl-shaped **C1SS** dimer was directly overlapped with a translational motion along the π-dimer with the sterically repulsive S–S interactions along the π-stack, a large relative energy around 30 kcal mol$^{-1}$ was observed as a forbidden in-plane free rotational motion. The energy minima for the skew angle rotation were observed at the angles of 30° and 90° with every 60° rotation of the upper **C1SS**, while the maxima of the dipole moment of 0.84 D were also observed around 20 and 100°. The forbidden in-plane free rotation inhibited the formation of Col$_h$ liquid crystal phase. The restricted 30°-rotated π-stacking interaction of **C1SS** in the π-stacking column was also thermally fluctuated in the high-temperature viscous S2 phase, where the motion was also coupled with the inversion motion of bowl-shaped conformation via the π-planar one (Fig. 8c). Although the π-stacking distance in the π-dimer (**C1SS**)$_2$ was associated with the relative energy, it did not affect the theoretical dipole moment (Supplementary Fig. 15). Three kinds of motional freedoms of torsion, skew, and dimer separation distance were activated at S2 phase, among which the first two coupled to generate the macroscale dipole moment under the applied electric field (Fig. 8d). When the electric field was applied to the thermally fluctuated π-stacking column of (**CnSS**)$_\infty$, the polarized molecular conformation in the 1D assembly should be stabilized at the bowl-shaped and twisted π-stacking columnar structure keeping the maximum dipole structure. Such polar molecular assembly was thermally relaxed after the removal of the outer electric field, which resulted in the ferroelectric $P$-$E$ hysteresis and relaxation behavior at high-temperature viscous S2 phase. In the order-disorder type molecular ferroelectrics, the low-frequency response at $f = 0.1$–$10$ Hz has been typically observed in the $P$-$E$ hysteresis curve. However, the relatively high-frequency responses of the $P$-$E$ hysteresis curve at $f = 100$–$200$ Hz for **CnSS** derivatives were consistent with the structural relaxation process from the polar molecular assembly structure under the applied electric field.

In summary, the collective dipole inversion of 1D π-stacking bowl-shaped molecules showed the ferroelectric response due to bowl-to-bowl inversion, where the chemical design of the bowl-depth (CH$_2$ → S → Se) tuned the suitable inversion barrier in bulk and possible dipole inversion by the application of the electric field. **CnSS** derivatives indicated the phase transition to high viscous S2 phase, where complete melting state of alkyl chains formed the 1D plastic crystalline phase. The thermally activated highly melting state enabled collective bowl-to-bowl inversion associated with the in-plane rotation stabilized by S•••S contacts and dipole-dipole interactions. The $P$-$E$ hysteresis responses were adequately high even in dynamic molecular system, caused by the high-speed relaxation of the polar bowl-stacking π-column under the electric field. The present finding reveals that the wide variation of molecular designs for organic π-molecules enables us to fabricate flexible and high-density organic memory devices.

## Methods

**Materials.** C6TP and C4SeS were prepared following reported methods[32,35]. **C2SS** was synthesized by a modified method for the synthesis of other trichalcogen-asumanene derivatives (Supplementary Information, section 2)[32]. **CnSS** ($n = 6, 8, 10,$ or $16$) were prepared by the following general procedure: a dichloromethane solution of boron tribromide (1.0 mol L$^{-1}$, 10 equiv.) was slowly added to a dichloromethane solution of **C2SS** at 0 °C, before the mixture was stirred at room temperature for 2 h. Then, water was slowly added to the reaction mixture at 0 °C. The resulting solid was collected by filtration and washed with dichloromethane and water to furnish intermediate **C0SS**. The obtained solid was dissolved in DMF, and potassium carbonate (4–10 equiv.) and the corresponding alkyl bromide (4–8 equiv.) were added to the solution. The mixture was stirred overnight at 75 °C. After cooling to room temperature, the mixture was filtered to remove an insoluble solid. Water was added to the obtained solution, and the organic layer was extracted with a mixture of ethyl acetate and hexane (4:1, v/v). The combined organic extracts were dried over anhydrous magnesium sulfate before the inorganic solids were removed by filtration. After evaporation of all volatiles, the crude products were purified by column chromatography on silica gel (eluent: hexane/dichloromethane) to afford **CnSS**.

**Measurements.** Temperature-dependent crystallographic data for a single crystal of **C2SS** at 300 K were collected on a Rigaku RAPID-II diffractometer equipped with a rotating anode and a multilayer confocal optic using graphite-monochromated Cu-Kα radiation ($\lambda = 1.54187$ Å). Structural refinements were carried out using the full-matrix least-squares method on $F^2$. Temperature-dependent PXRD measurements were performed on a Rigaku SmartLab diffractometer using Cu-Kα radiation ($\lambda = 1.54187$ Å). Temperature-dependent dielectric constants were determined by the two-probe AC impedance method at frequencies from 1 kHz to 1 MHz (Hewlett-Packard, HP4194A) with a liquid crystal cell placed in a temperature control system (Linkam, LTS350). The $P$–$E$ curves were measured with a commercially available ferroelectric tester (Precision LC, Radient Technologies) using a liquid crystalline cell with an electrode gap of 2 μm. The **CnSS** derivatives in the liquid state were introduced in the electrode gap of the liquid crystal cell.

## Data availability

All the data generated or analyzed during this study are included in this published article (and its supplementary information files) or available from the authors upon reasonable request. The crystallographic information for this paper has been deposited at The Cambridge Crystallographic Data Center (http://www.ccdc.cam.ac.uk) under reference number CCDC-1912824.

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

## Acknowledgements

This work was supported by a Grant-in-Aid for Scientific Research on Innovative Areas "π-System Figuration: Control of Electron and Structural Dynamism for Innovative Functions" from the Japan Society for the Promotion of Science (JSPS) (JP26102006 to M.S. and JP26102007 to T.A.) and "Stimuli-responsive Chemical Species for the Creation of Functional Molecules" (15H00918 to S.F.) from the Japanese Ministry of Education, Culture, Sports, Science, and Technology (MEXT). This work was also supported by JSPS KAKENHI grant (16K13945 to S.F., JP20H05865 to T.A., JP20K05442 and JP20H04655 to T.T.), JST CREST (JPMJCR18I4 to T.A.), and the 'Dynamic Alliance for Open Innovation Bridging Human, Environment and Materials' from MEXT (T.A.). M.S. gratefully acknowledges a research grant from the Yamada Science Foundation.

## Author contributions

M.K. and K.H. conducted the synthetic experiments and analyzed the data. J.W. performed thermal and electrical measurements. N.H. carried out the single-crystal X-ray diffraction analysis. T.T. performed DFT calculations. J.K. and Y.S. conducted SHG measurement. S.F., M.S. and T.A. prepared the manuscript. S.F. and T.A. conceived the project, which was directed by S.F., M.S. and T.A.

## Competing interests

The authors declare no competing interests.
