## [Peer Review File · Nature Communications]

Reviewers' comments:

Reviewer #1 (Remarks to the Author):

The manuscript "Ferroelectric Crystalline Columnar Assemblies for the Bowl-to-Bowl Inversion of Aromatic Cores" outlines some excellent work synthesizing and characterizing fused-ring bowl-shaped heteroaromatic molecules, including x-ray diffraction, temperature-dependent ferroelectric measurements, and synthesis of related compounds.

I think my big concern is whether this is truly a "new concept" in ferroelectrics, particularly given the long tradition of bowl-shaped ferroelectrics the authors themselves indicate in Figure 1. I'll come back to that.

On a technical level, I think the paper is very strong. These are difficult molecules to synthesize, and the concept of a sulfur-substituted core to decrease the bowl depth is very interesting. The result is a much lower activation energy for interconversion. This much is well understood.

What I'm not sure I understand is how the ferroelectric CnSS compare to the selenium counterpart CnSeS. If this is a truly "new concept" then the presence of a bowl-like solid state for CnSeS compounds should also yield ferroelectric response. No, C4SeS is a rigid and 'supresses the thermally-activated bowl-to-bowl inversion.' Okay, but then a reader is left trying to find related compounds to CnSS that have very specific solid-state organization?

My main point is that the manuscript itself indicates that bowl-shaped ferroelectric compounds have been known for a while. This particular family is very interesting (esp. since the hydrocarbon variants don't show this behavior) and the work is well done. But I'm not sure the impact since the obvious relative (selenium) does not show response.

One minor technical comment - the computed dipole moments don't mean much. The ferroelectric response occurs in a column .. since the molecules are highly conjugated, they likely have high polarizability. i would think a better option is to take the crystal structure, take at least 2 molecules, and compute the response of that assembly (i.e., accounting for the polarization of neighbors).

In principal, the polarization times the applied field should give you a potential energy term.. I would be very curious to see if the thermal energy plus that computed potential energy term is sufficient to overcome the activation barrier. Presumably it is, since the inversion occurs, but it's possible that there's some sort of cooperative effect in the solid state, which would be useful to note.

Reviewer #2 (Remarks to the Author):

The paper describes a study of bowl-shaped molecular dipoles that can stack and crystallize in polar structures. Detailed measurements of crystal structure, dielectric response, calorimetry, and polarization hysteresis provide strong evidence that some of the compounds are ferroelectric. The most convincing candidates are the ones that are properly crystalline with polar symmetry and

evidence of polarization reversal.

The manuscript is well organized and clearly written. The figures are clear and well-labeled, except as noted below. The materials and methods information is adequate guiding reproduction of the work by scientists of comparable skill. The results are novel and of potential interest in the field of molecular ferroelectric materials, as well as allied fields of molecular crystal engineering, organic electronics, and structural chemistry. I recommend publication in Nature Communications after the authors have made improvements addressing the following comments.

The main results should be organized in a comprehensive table listing, for example, compositions, structural symmetries (noting whether or not there is single-crystal data), dipole moments, phases, phase transition and melting temperatures, polarization and coercive field measurements, and perhaps other features bearing on the question of whether or not the data adequately support the existence of ferroelectric state for each compound. Comment on any trends in reference to this table.

Regarding Equation (1), it should be noted that molecular ferroelectric materials generally do have spontaneous polarizations larger than individual the dipole density due to the collective enhancement that stabilizes ferroelectricity. Recent theoretical work using first-principles computational modeling have explained this clearly. I recommend referring to such studies reported by Nakhmanson, Xiong, and Horiuchi, for example.

In addition, I recommend the following minor corrections.

The direction of temperature change should be clearly indicated in the in figure 3a. The directions of the hysteresis loops should be indicated in figures 5c and 6. The same indications should be added to the corresponding figures in the supplementary material.

In line 28, change "scare" to "scarce".

In line 34, change "conducting" to "conduction"

Reviewer #3 (Remarks to the Author):

This article by Furukawa et al. represents the first example (as the authors suggest and also to the best of my knowledge, based on my experience and a literature survey) of "Columnar (liquid) Crystalline Assemblies that show Ferroelectricity due to Molecular Bowl Inversion". However, all words in this sentence are necessary to define the unprecedented nature of this work. For instance:

- 1 Bowl-shaped assemblies in liquid crystalline phases have already shown excellent ferroelectric properties due to bowl inversion (Science 336, 209 (2012); cited and shown in Fig. 1)
- 2 Related bowl-shaped molecules like corannulene have shown uniaxial alignment in the presence of electric fields (JACS, 131, 44 (2009); cited) or (JACS 133, 13767 (2011); not cited)
- 3 Bowl-shaped molecules with strong axial dipoles have shown permanent homeotropic alignment and polarization in columnar liquid crystals after oriented with electric fields (Chem. Mater. 3, 985

(2015); not cited) and ferroelectric properties in nematic phases (Adv. Mater. 27, 4280 (2015); not cited). However, the latter effect is due to reorientation of the whole self-assembled columnar structures.

I believe the use of oligo-(vinylidenedifluoride) and/or amide side chains should be mentioned as an alternative way to induce ferroelectricity in columnar organic p-conjugated materials (JACS, 138(19), 6217 (2016); not cited) and (Sci. Adv. 3, e1701017 (2017); not cited). The latter shows also an example of a columnar ferroelectric mesophases from bowl-shaped molecules, but polarization mainly arises from the orientation of the peripheral dipolar amide groups

So, these new references are very relevant and should be cited in the text. In any case, the unprecedented nature of the work is maintained: this is the first time ferroelectricity is demonstrated to arise as a function of bowl inversion in a molecule. The article is also well written and the experiments carefully performed and described in sufficient detail. The conclusions are adequately supported by experimental evidence. For these reasons I would recommend publication after considering a few major issues that should be addressed:

1. Are the authors able to align their materials at high temperatures in the presence of electric fields? Uniaxial homeotropic alignment should be demonstrated if that is the case. However, as the authors indicate, most of their samples are non-fluid. In that case, I guess the material is composed of different domains with different column orientations and the maximum attainable polarization cannot be reached (which, by the way, is not extraordinarily high). Could the authors comment on this or, otherwise, demonstrate homeotropic alignment with E-fields in Linkam cells with transparent ITO electrodes?

2. A second comment I have is on the dependence of the polarization with time after the material is aligned and the electric field is switched off, which is essential for practical applications. Does the material retain the polarization? If so, For how long? What is then the response with time? A key technique useful for both points is SHG, that would prove the non-centrosymmetrical nature of the material in the presence/absence of fields.

Finally, a few typos:

Page 7, 177: "T- and f- independent" Is this right? Or should it be "-dependent"?

Page 8, 184: Fig. 3c should be Fig. 5c

Page 8, 205: "serves as a heat bath" is probably not the best expression

Overview of our revisions

The overall changes and conclusions from additional experiments are described below.

We conducted several experiments to reconsider our findings based on reviewers' suggestions including a polarization magnitude, SHG activities, theoretical calculations using model compounds, and polarization retention times. It took time for the revision, since we resynthesized all the titled molecules and carried out the above-mentioned measurements. Three new co-authors were added in the manuscript for the measurements.

• SHG activities of C_nSS derivatives

The important point to mention is that **C2SS**, whose molecular structure is clarified by the single crystal X-ray structure analysis, showed the SHG activity derived from the polar space group. The sample of **C2SS** was crystalline solid at room temperature, whereas the other derivatives with long alkyl chains (C_nSS , $n = 6, 8, 10, 16$) were viscous solids at room temperature. We confirmed by PXRD analysis that the alkyl chains of these derivatives are considerably melted and thermally activated molecular motions exist at high temperatures (Fig. 4a). In the measurements of SHG activity of the long alkyl-chain derivatives at S2 phase, we could not observe reproducible SHG responses. Although we sometimes observed the SHG activity at the measurement of high crystalline domains, the intensity of the SHG signals at S2 phases were almost the same as those of S1 phases. It should be noted that the titled molecules respond at relatively high frequencies in the P - E hysteresis curves, whereas the conventional order-disorder organic ferroelectrics show P - E hysteresis at low frequencies. On the basis of these experimental results, we discussed a plausible mechanism about the ferroelectric P - E hysteresis by theoretical calculations of a model compound **C1SS**.

• The mechanism

We concluded that the ferroelectric P - E hysteretic responses are induced by dielectric relaxations coupled with the unique bowl shape and dipole structures of the C_nSS . We estimated the energy difference between the planar and bowl-shaped structures of the model compound **C1SS**. The theoretical calculations revealed that structural fluctuation between the bowl and planer structures in **C1SS** occurs even at room temperature. This conformational change compensates the dipole moment of C_nSS molecules, which is consistent with the experimental fact of the disappearance of SHG activity. In addition, we estimated the rotational barriers of **C1SS** in the π -stacked column in a π -stacked dimer model. The dimer showed high rotational barriers because of intermolecular $S\cdots S$ interactions. In the high temperature S2 phases, the π - π stacking, the bowl inversions, and the restricted in-plane rotation are considered to be thermally activated. By applying an electric field to the thermally-activated-state, the

molecules could deform into a bowl shape and reassemble to maximize the macroscopic dipole moment through the change of the skew angles and the π -stack distances in the π -stacking columns. In general paraelectrics, the macroscopic dipole moment instantaneously relaxes without applying field. In contrast, the relaxation of the inversion-rotation motion of **CnSS** was reflected in relatively slow frequency at about 100-200 Hz, which resulted in hysteresis in the *P-E* curves. This interpretation is also consistent with the hysteresis curves in the *P-E* plots at the relatively fast frequencies.

We added these discussions mentioned above in the final section of the main manuscript, and greatly appreciate reviewers' valuable comments. Our responses to each of reviewers' comments are described below. The changes in the manuscript and Supporting Information are highlighted in yellow color.

Green: Reviewers' comments

Black: Our responses

Reply to Reviewer #1's comments

The manuscript "Ferroelectric Crystalline Columnar Assemblies for the Bowl-to-Bowl Inversion of Aromatic Cores" outlines some excellent work synthesizing and characterizing fused-ring bowl-shaped heteroaromatic molecules, including x-ray diffraction, temperature-dependent ferroelectric measurements, and synthesis of related compounds.

【Comment 1-1】

I think my big concern is whether this is truly a "new concept" in ferroelectrics, particularly given the long tradition of bowl-shaped ferroelectrics the authors themselves indicate in Figure 1. I'll come back to that.

Our response:

We used "a new concept" as a concept of invertible aromatic cores rather than "bowl-shaped" ferroelectrics. To avoid such misunderstanding, we changed the sentence "a new concept" to "a concept of invertible aromatic cores", and added appropriate references suggested by reviewer #3 to clarify the unprecedented findings in this work.

【Comment 1-2】

What I'm not sure I understand is how the ferroelectric CnSS compare to the selenium counterpart CnSeS. If this is a truly "new concept" then the presence of a bowl-like solid state for CnSeS compounds should also yield ferroelectric response. No, C4SeS is a rigid and 'suppresses the thermally-activated bowl-to-bowl inversion.' Okay, but then a reader is left trying to find related compounds to CnSS that have very specific solid-state organization?

My main point is that the manuscript itself indicates that bowl-shaped ferroelectric compounds have been known for a while. This particular family is very interesting (esp. since the hydrocarbon variants don't show this behavior) and the work is well done. But I'm not sure the impact since the obvious relative (selenium) does not show response.

Our response:

Important factors of the bowl inversion under applied external electric fields are caused by the small inversion barrier of the π -aromatic core and the entropy source derived from the melting of the alkyl-chains. The melting of the side alkyl-chains increases an entropy at high temperatures. Thus increased entropy (heat bath) is essential to bring about the bowl inversion in the solid state. The selenium derivative **C4SeS** did not show the bowl inversion because of lack of S2 phase originating from the melting of the alkyl-chains (P.11 in manuscript, Supplementary Figure 12).

【Comment 1-3】

One minor technical comment - the computed dipole moments don't mean much. The ferroelectric response occurs in a column .. since the molecules are highly conjugated, they likely have high polarizability. I would think a better option is to take the crystal structure, take at least 2 molecules, and compute the response of that assembly (i.e., accounting for the polarization of neighbors).

In principal, the polarization times the applied field should give you a potential energy term. I would be very curious to see if the thermal energy plus that computed potential energy term is sufficient to overcome the activation barrier. Presumably it is, since the inversion occurs, but it's possible that there's some sort of cooperative effect in the solid state, which would be useful to note.

Our response:

According to Reviewer#1's suggestion, we calculated changes of dipole moments and relative energies of a dimer of model compound **C1SS**. We added the experimental results (Fig. 7) and considerations in the last section of the manuscript. The important point is that **CnSS** can take the bowl-shaped structure and skew angles to maximize dipole moments under the external electric fields at high temperature region such as S2 phase.

Reply to Reviewer #2's comments

The paper describes a study of bowl-shaped molecular dipoles that can stack and crystallize in polar structures. Detailed measurements of crystal structure, dielectric response, calorimetry, and polarization hysteresis provide strong evidence that some of the compounds are ferroelectric. The most convincing

candidates are the ones that are properly crystalline with polar symmetry and evidence of polarization reversal.

The manuscript is well organized and clearly written. The figures are clear and well-labeled, except as noted below. The materials and methods information is adequate guiding reproduction of the work by scientists of comparable skill. The results are novel and of potential interest in the field of molecular ferroelectric materials, as well as allied fields of molecular crystal engineering, organic electronics, and structural chemistry. I recommend publication in Nature Communications after the authors have made improvements addressing the following comments.

【Comment 2-1】

The main results should be organized in a comprehensive table listing, for example, compositions, structural symmetries (noting whether or not there is single-crystal data), dipole moments, phases, phase transition and melting temperatures, polarization and coercive field measurements, and perhaps other features bearing on the question of whether or not the data adequately support the existence of ferroelectric state for each compound. Comment on any trends in reference to this table.

Our response:

According to Reviewer #2's suggestion, we added the summary table in the main manuscript (Table 1).

【Comment 2-2】

Regarding Equation (1), it should be noted that molecular ferroelectric materials generally do have spontaneous polarizations larger than individual the dipole density due to the collective enhancement that stabilizes ferroelectricity. Recent theoretical work using first-principles computational modeling have explained this clearly. I recommend referring to such studies reported by Nakhmanson, Xiong, and Horiuchi, for example.

Our response:

According to the reviewer's suggestion, we added an explanation of the theoretical studies and appropriate references by Nakhmanson, S. M. et al (Polarization canting in ferroelectric diisopropylammonium-halide molecular crystals: a computational first principles study. *J. Mater. Chem. C* **6**, 1143–1152 (2018)) and Horiuchi, S. et al (Proton tautomerism for strong polarization switching. *Nat. Commun.* **8**, 14426(2017)) in the manuscript (#33 and #34).

【Comment 2-3】

In addition, I recommend the following minor corrections.

The direction of temperature change should be clearly indicated in the in figure 3a. The directions of the

hysteresis loops should be indicated in figures 5c and 6. The same indications should be added to the corresponding figures in the supplementary material.

In line 28, change “scare” to “scarce”.

In line 34, change “conducting” to “conduction”

Our response:

We added arrows to show the directions of the hysteresis loops in figure 5c and 6, and corrected the typos.

Reply to Reviewer #3's comments

【Comment 3-1】

This article by Furukawa et al. represents the first example (as the authors suggest and also to the best of my knowledge, based on my experience and a literature survey) of “Columnar (liquid) Crystalline Assemblies that show Ferroelectricity due to Molecular Bowl Inversion”. However, all words in this sentence are necessary to define the unprecedented nature of this work. For instance:

1. Bowl-shaped assemblies in liquid crystalline phases have already shown excellent ferroelectric properties due to bowl inversion (Science 336, 209 (2012); cited and shown in Fig. 1)
2. Related bowl-shaped molecules like corannulene have shown uniaxial alignment in the presence of electric fields (JACS, 131, 44 (2009); cited) or (JACS 133, 13767 (2011); not cited)
3. Bowl-shaped molecules with strong axial dipoles have shown permanent homeotropic alignment and polarization in columnar liquid crystals after oriented with electric fields (Chem. Mater. 3, 985 (2015); not cited) and ferroelectric properties in nematic phases (Adv. Mater. 27, 4280 (2015); not cited). However, the latter effect is due to reorientation of the whole self-assembled columnar structures.
4. I believe the use of oligo-(vinylidenedifluoride) and/or amide side chains should be mentioned as an alternative way to induce ferroelectricity in columnar organic p-conjugated materials (JACS, 138(19), 6217 (2016); not cited) and (Sci. Adv. 3, e1701017 (2017); not cited). The latter shows also an example of a columnar ferroelectric mesophases from bowl-shaped molecules, but polarization mainly arises from the orientation of the peripheral dipolar amide groups.

So, these new references are very relevant and should be cited in the text. In any case, the unprecedented nature of the work is maintained: this is the first time ferroelectricity is demonstrated to arise as a function of bowl inversion in a molecule. The article is also well written and the experiments carefully performed and described in sufficient detail. The conclusions are adequately supported by experimental evidence.

Our response:

We appreciate Reviewer#3's valuable suggestions. As Reviewer#1 also mentioned, we revised the

sentence “a new concept” to “a concept of invertible aromatic cores” to clarify a novelty of this work. The suggested references were also added in the manuscript with explanations of how the present molecules are unique in the bowl-shaped molecules/assemblies.

【Comment 3-2】

Are the authors able to align their materials at high temperatures in the presence of electric fields? Uniaxial homeotropic alignment should be demonstrated if that is the case. However, as the authors indicate, most of their samples are non-fluid. In that case, I guess the material is composed of different domains with different column orientations and the maximum attainable polarization cannot be reached (which, by the way, is not extraordinarily high). Could the authors comment on this or, otherwise, demonstrate homeotropic alignment with E-fields in Linkam cells with transparent ITO electrodes?

Our response:

Heated liquid phase samples were inserted into the sandwich electrode under N₂ atmosphere, and slow cooling from liquid to solid state under the DC electric field aligned the π -stacked column along the direction normal to the ITO surfaces. However, almost the same polarization magnitude was observed for poling and non-poling treatments. The **CnSS** itself tends to form homeotropic orientation of π -stacked column during the cooling process from liquid to solid. That point was simply mentioned in the text and new dark POM images to evidence the homeotropic orientation were newly included in Supporting Information Figure S11.

【Comment 3-3】

A second comment I have is on the dependence of the polarization with time after the material is aligned and the electric field is switched off, which is essential for practical applications. Does the material retain the polarization? If so, For how long? What is then the response with time?

A key technique useful for both points is SHG, that would prove the non-centrosymmetrical nature of the material in the presence/absence of fields.

Our response:

Unfortunately, we could not discuss the polarization considering symmetry of the crystals, because SHG signal was not clearly observed in most of the **CnSS** derivatives except for **C2SS**. We measured time-dependent polarization behavior of **C10SS** at 343 K, and it showed that the polarization was retained for 4000 ms after applying a pulse voltage. We added this result in the manuscript and Supplementary Figure 9.

【Comment 3-4】

Finally, a few typos:

Page 7, 177: “T- and f- independent” Is this right? Or should it be “-dependent”?

Page 8, 184: Fig. 3c should be Fig. 5c

Page 8, 205: “**serves as a heat bath**” is probably not the best expression

Our response:

We corrected the typos and changed the sentence “a heat bath” to “an entropy source”.

REVIEWERS' COMMENTS

Reviewer #2 (Remarks to the Author):

I am satisfied by the revised manuscript and supplementary materials and support publication of the paper in Nature Communications.

Reviewer #3 (Remarks to the Author):

I am mostly satisfied with the new experiments performed by the authors and the modifications made to the text and supporting information. I would have liked to see more about the time-dependent evolution of the polarization but unfortunately SHG responses could not be clearly observed

I believe this nice and relevant work can be published as it is now